# LoCHAid: An ultra-low-cost hearing aid for age-related hearing loss

Soham Sinha[1¤], Urvaksh D. Irani[2], Vinaya Manchaiah[3], M. Saad Bhamla[1]*

**1** School of Chemical & Biomolecular Engineering, Georgia Institute of Technology, Atlanta, GA, United States of America, **2** School of Mechanical Engineering, Georgia Institute of Technology, Atlanta, GA, United States of America, **3** Department of Speech and Hearing Sciences, Lamar University, Beaumont, TX, United States of America

¤ Current address: School of Bioengineering, Stanford University, Palo Alto, CA, United States of America
* saadb@chbe.gatech.edu

**Data Availability Statement:** All data is available on GitHub https://github.com/bhamla-lab/LoCHAid-2020-PLOS-ONE.

**Funding:** M.S.B. acknowledges funding support from Capita Foundation (2018 CFAR award).

## Abstract

Hearing aids are the primary tool in non-medical rehabilitation for individuals with hearing loss. While the costs of the electronic components have reduced substantially, the cost of a hearing aid has risen steadily to the point that it has become unaffordable for the majority of the population with Age-Related Hearing Loss (ARHL) especially for those residing in low- and middle-income countries. Here, we present an ultra-low-cost, affordable and accessible hearing aid device ('LoCHAid'), specifically targeted towards treating ARHL in elderly patients. The LoCHAid components cost 98 cents (< $1) when purchased in bulk for 10,000 units and can be personalized for each user through a 3D-printable case. It is designed to be an over-the-counter (OTC) self-serviceable solution for elderly individuals with ARHL. Electroacoustic measurements show that the device meets most of the targets set out by the WHO Preferred Product Profile and Consumer Technology Association for hearing aids. The frequency response of the hearing aid shows selectable gain in the range of 4-8 kHz, and mild to moderate gain between 200-1000 Hz, and shows very limited total distortion (1%). Simulated gain measurements show that the LoCHAid is well fitted to a range of ARHL profiles for males and females between the ages of 60-79 years. Overall, the measurements show that the device offers the potential to benefit individuals with ARHL. Thus, our proposed design has the potential to address the challenge of affordable and accessible hearing technology for hearing impaired elderly individuals especially in low- and middle-income countries.

## Introduction

Age-Related Hearing Loss (ARHL) is one of the most prevalent chronic conditions in older adults with an estimated affected population of 226 million individuals over the age of 65 around the world, which is projected to grow to 900 million by 2050 [1]. Countries in sub-Saharan Africa, South Asia, and Asia Pacific have a prevalence of ARHL that is 4 times higher than in developed nations [1]. Although individual configurations may differ based on

http://www.capitafoundation.org and National Science Foundation. 2020 CAREER Award. The funders had no role in study design, data collection and analysis, decision to publish, or preparation of the manuscript.

**Competing interests:** The authors have declared that no competing interests exist.

location and other underlying etiologies, the condition is typically characterised by increasing hearing loss from 1 kHz onwards in the high frequency region [2, 3]. ARHL results in various physical, mental, and social consequences such as communication difficulties [4–6]. These combined can further exacerbate or cause anxiety, depression, and social isolation, leading to an overall lower health-related quality of life (HRQoL) [5, 7]. While there is no cure for ARHL, hearing aids are the primary and the most frequent tool used to rehabilitate individuals and improve their respective HRQoL. However, the adoption of hearing aids is very low amongst adults. In low- and middle-income countries (LMIC), hearing aid adoption rates are below 3% whereas in non-LMIC countries, the adoption rate is around 20% [1]. Various reasons (e.g., lack of self-awareness of hearing loss, unequal access to hearing healthcare) may contribute to this poor uptake [8]; however, cost is one of the most substantial factors [8–14]. The retail price of a pair of hearing aids range between $1,000 (low-end) to $8,000 (high-end), with an average price being $4,700 in the United States [7, 15].

The reasons for the high cost include proprietary software and hardware, costs of distribution, and the refusal of coverage by public policy programs like Medicare and private insurance companies [7, 9–11]. Even though various low-cost solutions ($< \$300$) have been developed in the last decade such as personal sound amplifiers (PSAPs), the majority of such devices have poor acoustic characteristics and do not meet the acoustic characteristics needed to treat ARHL. They are characterised by having too much low frequency gain and limited high frequency gain, dangerous levels of amplification, excessive internal noise, and high distortion [11, 13, 14, 16–19]. Moreover, over-the-counter (OTC) hearing aids and PSAPs are still between $100 and $500 [11, 14, 17–19], which is significantly expensive for people living in LMIC, where the annual healthcare expenditure per capita ranges from $5 to $50 (2010 USD) [20, 21]. Thus, there is an urgent global need for accessible and affordable hearing devices, potentially served OTC similar to reading glasses, which is further advocated by both the World Health Organisation and the U.S. National Academies of Science and Engineering [1, 12, 22].

To address this need, we explore the development of a minimal component hearing aid to address ARHL. We aim to engineer an accessible and affordable minimal device with the required electroacoustic characteristics to benefit elderly users with ARHL. To that extent, we develop a hearing device, coined 'LoCHAid', which costs $0.98 in components (inlcuding batteries) for mass-production of 10,000 units, excluding labor costs. We test the device in laboratory conditions using two methods. First, we test the electroacoustic characteristics in an anechoic chamber to examine its properties such as gain, frequency response, harmonic distortion, and equivalent input noise. Second, we simulate the preferred gain for a range of ARHL profiles (S1 Fig in S1 File) in a coupler using the Audioscan Verifit device, and through a real-ear simulator using the G.R.A.S KEMAR manikin. We compare the LoCHAid response to these profiles and show that the device provides appropriate gain for a range of average mild to moderate ARHL audiometric patterns, for both males and females (left and right ears) in the age range of 60-79 years.

## Methods

### Construction of LoCHAid

The LoCHAid was constructed using a handheld soldering iron (X-Tronic Model 3020-XTS Digital Display Soldering Iron Station) with solder (WYCTIN 1.0mm 50G 60/40 (Tin-60% Lead-40%) Tin Lead Roll 1.8% Flux Soldering Wire Reel). Foam (EVA Straight Edge Foam) was obtained for ease of construction for the microphone placement, but can be removed after construction (S1 Video). Construction takes 30 minutes (S1 Video). The case was designed in

SolidWorks v27, and was 3D-printed (Stratasys J750) from blue polyamide (Nylon 12). The electret microphone utilisng MAX 9814, class D stereo amplifier utilising MAX 98306, audio jack, coin cell holder, and 3V coin cell battery was obtained from Adafruit (www.adafruit.com, P/N 1713, 987, 1699, 1870, 2849, respectively). The 5kΩ resistor, 1uF capacitor, 6.8kΩ resistor, 1000pF capacitor, 15uF capacitor, 6 pronged on/off slide switch, and was obtained from Digikey (www.digikey.com, P/N CT6EP502, C0805C105J4RACTU, RMCF0805FT6K80, CL21B102KBANNNC, C1210C156K8PACTU, JS202011CQN, respectively). The potentiometer which provides a volume control of (+/- 10 dB SPL) was obtained from Amazon (www.amazon.com, P/N MCIGICM Potentiometer Breadboard Kit with Knob). The circuit board was printed at Oshpark board printing services (www.oshpark.com). One of the filters is a second-order high-pass RC filter with a cutoff frequency of 2340 Hz (constructed of 2 6.8kΩ, and 2 1000pF capacitors). The other is a low-pass powerline filter to subdue noise from the power source. The power source range is 3V–5.5V. The schematic is shown in S6 Fig in S1 File.

## Electroacoustic analysis

Electroacoustic measurements were performed using the AudioScan Verifit device (version 3.1; AudioScan, Dorchester, ON, Canada). For all tests, a pair of Panasonic RP-HJE125E Wired Earphones—Wired, Orange (RP-HJE125-D) was used. The earbuds' soft plastic bud was removed, and the exposed end was placed into the center of a HA-1 0.2 cc-coupler. Putty (Scotch Lightweight Mounting Putty, 2 oz) was used to seal the coupler, and any other sound openings of the earbud itself outside the coupler. The device was placed inside the anechoic chamber of the machine. The other earbud was sealed off to prevent feedback (S2 Video). The AudioScan speaker was placed within 2mm of the microphone of the device. The entire chamber was completely closed, and the tests were run. The measurements obtained from the LoCHAid were compared against two hearing aid standards, including: (a) WHO preferred product profile for hearing aid technology suitable for LMIC [12]; (b) ANSI S3.22–2014/CTA-2051 standards for OTC devices [23, 24]. However, considering that the device is primarily aimed towards ARHL individuals in LMIC, the WHO specifications were used for most of the comparisons. The measurements included: output sound pressure level-90 (OSPL-90) curves, high-frequency average full-on gain (HFA FOG), frequency response curves, equivalent input noise (EIN), and total harmonic distortion (THD). All tests were run with 3 different devices, N = 3, with n = 15 trials per device.

## Simulated gain measurements against ARHL profiles

The preferred gains for a range of mild- to moderate ARHL profiles (see S1 Fig in S1 File) were simulated and were compared the LoCHAid response to these profiles to check if the device provides appropriate gain for certain ARHL audiometric patterns. The simulated gain measurements were performed using two different methods, which included: (a) Speechmap testing simulating hearing aid gain in a coupler; and (b) simulation in an ear simulator using the KEMAR manikin. The type and extent of ARHL varies across age, ear, and gender. Hence, a range of ARHL profiles was taken from published studies [2, 3] and the preferred gain was estimated using the NAL-NL2 prescriptive formula for these profiles. The speechmap and also KEMAR ear simulated measurements of the LoCHAid were compared against these preferred estimated gains. This comparison was to determine whether the LoCHAid could provide appropriate levels of amplification (within 5 or 10 dB SPL) at 10 frequencies (250, 500, 750, 1000, 1500, 2000, 3000, 4000, 6000, 8000 Hz).

The Speechmap test was performed using the AudioScan Verifit device. The 0.2 cc-coupler was switched out with a 0.4 cc wideband coupler; the same procedure was followed with

removing soft plastic earbuds, and placing the bare plastic part in the middle of the coupler, and sealing the entering side of the coupler. Other holes were also sealed off. The entire chamber was closed and then the tests were run using the ISTS (International Speech Test Signal). ISTS is an internationally recognized test signal that may be used in the technical evaluation of hearing instruments, and for probe-microphone measurements [25]. The ISTS is shaped according to the LTASS (Long Term Average Speech Spectrum) standards. Three test signal strengths were run at 55 dB SPL (soft/whispering), 65 dB SPL (average/conversational), and 80 dB SPL (loud/outside). All tests were run with N = 3 devices, n = 15 trials overall.

The simulated real-ear feedback measurements were conducted in the G.R.A.S KEMAR manikin's left ear. The tests were conducted in an audiological soundproof room with the manikin inside. The LoCHAid was clipped to the front of the manikin's shirt. The earphones (Panasonic RP-HJE125E Wired Earphones) were placed inside the manikin's ears with the soft plastic buds attached. The loudspeaker was located at an azimuth of 45 degrees and 30 cm (1 foot) from KEMAR. The center of the loudspeaker was at the same level as the midpoint of the hearing aid. To simulate a normal conversational situation, the input signal used was ISTS at 65 dB SPL. A single device was tested by playing the exact 40 seconds of the recording. The experimental setup was re-calibrated after every run to make sure that the intensity of the incoming sound was still at 65 dB SPL, and earbuds if they slipped out were placed back in the ears. The test was run with N = 1 device, n = 15 trials.

## Results

### LoCHAid as a modular device

The LoCHAid is a modular hearing aid device, which is based primarily on mass manufactured modular components. These include an electret microphone with an automatic gain control and preamplifier, a Class D Stereo Amplifier, a frequency filter, and a standard 3.5 mm audio jack. The audio jack allows for direct audio output and it allows the use of any closed form sound transducer such as headphones, or earphones. The frequency filter is a second-order high-pass passive resistor-capacitor (RC) filter with a cutoff frequency of 2340 Hz, which enables shaping the response curve. Peripherals such as an on/off switch, volume control knob (potentiometer), and a power source input are included and shown in Fig 1c and 1e. The power source requirement is small (3-5.5 V) and can be provided from varied a variety of sources such as rechargeable AAA, AA, coin cell, and rectangular lithium ion batteries as shown in Fig 1b. To protect against noise from the power source, a low-pass DC powerline filter is used. For the most compact version, the lithium ion coin cell battery is used (Fig 1a).

To create the device, the components are soldered on to a custom printed circuit board (Fig 1c and 1d). The schematic for the board is shown in S6 Fig in S1 File. The board requires few soldering points and the entire device can be created in under 30 minutes with a soldering iron (S1 Video). To compactly hold and protect the LoCHAid, a self-fitting 3D-printed case was constructed from polyamide (Nylon 12) (Fig 1a). The configuration is body-worn with attached headphones. However, the device can also be placed in pockets or worn on the arm (Fig 1f). An end user can turn the device on and off, remove the case, replace batteries, turn the volume control knob, and attach headphones.

The device is designed to be durable. The LoCHAid is drop-proof from 6 ft over repeated impacts (12x) and water-proof up to 6 cm of depth for 15 seconds (S3 and S4 Videos and S5 Fig in S1 File). The recurring costs of batteries is mitigated by the low-power consumption of the device. It can function approximately 72 hours continuously with a single cell lithium ion battery, or a maximum of 21 days continuously with 2 AA batteries with an average background sound input of 55-60 dB SPL. Although costs of batteries may differ from country to

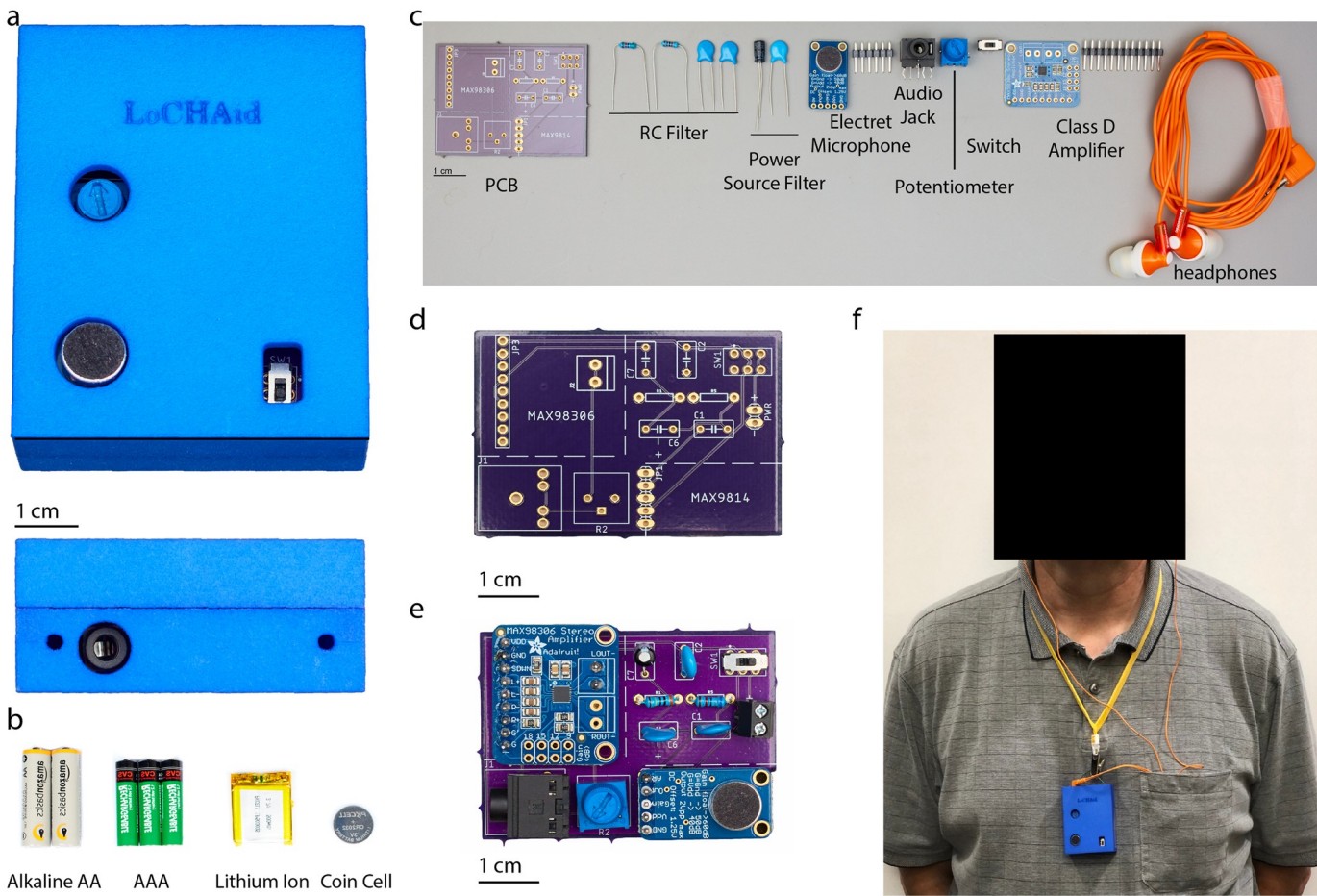

**Fig 1. Construction and components of the LoCHAid. a**. The LoCHAid is shown in its top view, with its 3D printed polyamide (Nylon 12) case tilted. The side view of the audio jack opening and holes for attaching material for neck wear are shown below. The LoCHAid in its case has a size of 6.70 mm by 5.70 mm. The audio jack can incorporate any standard 8 mm sound transducer. **b**. Displays various types of batteries such as AA, rechargeable AAA, Lithium Ion flat pack, as well as lithium ion coin cell that can be used to power the device. The device has a power requirement that is between 3-5.5 V. The amount of batteries denote the the number required to power the device. **c**. The required parts to assemble the device are shown here with group labels; specific details are given in Table 1. **d**. View of the custom printed circuit board (PCB) without any components. **e**. View of the PCB with components soldered on. **f**. View of the body-worn device by an anonymous 65 year old male as part of the intended audience of the device.

country, the cheapest per month replacement cost is associated with AA batteries compared to Lithium ion batteries. Additionally, AA batteries are more widely accessible. The operating temperature range is from -25°C to 65°C. The lifespan of the device is estimated to be 1.5 years.

LoCHAid does not over-amplify loud sounds. There is an inbuilt safety mechanism (Automatic Gain Control) if the input sound goes above 110 dB SPL; the device employs an attack and compression ratio of 1:500, and the sound is compressed to below 110 dB SPL after a hold time of 30 ms [26]. As a result of the hold time, small interval sharp sounds such as vehicle horns (100-120 dB SPL) are effectively protected against. To diminish loud continuous sounds such as rock concert music (100-130 dB SPL), a user can reduce the amplification easily using the in-built volume control. The gain of the LoCHAid is nonlinear as a result of the automatic gain control system of the MAX9814 component [26]. However, since the frequency control is fixed through a single channel in the MAX98306 amplifier circuit, the output curve shape is fixed across different volume configurations.

## Cost of manufacturing the LoCHAid

When mass produced at 10,000 units with earphones, a coin-cell battery and a holder, the LoCHAid has a cost of $0.98 (Table 1) considering all components are bought from the listed suppliers in Table 1 description. Since the LoCHAid is constructed out of mass produced open source electronics, it does not require specialty made parts. As a result, repairs can be completed by a minimally skilled user with a soldering iron and solder. Moreover, the low cost

Table 1. Component costs of the LoCHAid.

| Components | Mass Production Cost | |
|---|---|---|
| Earphones (**i**) | $0.04 | |
| Audio Jack (**ii**) | $0.03 | |
| 2 x 1000 pF Capacitor (**iii**) | $0.02 | |
| 2 x 1 uF Capacitors (**iv**) | $0.02 | |
| 1 x 15 uF Capacitor (**v**) | $0.01 | |
| 5 kΩ Trim Pot Potentiometer (**vi**) | $0.06 | |
| 6 pronged—Slide Switch (**vii**) | $0.03 | |
| Open Source Electret Microphone (**viii**) | $0.10 | |
| Open Source Stereo Class D 3.7 Amplifier (**ix**) | $0.48 | |
| Circuit Board (**x**) | $0.05 | |
| 3D Printed PLA Casing (**xi**) | $0.06 | |
| 2 x 6.8 kΩ Resistors (**xii**) | $0.02 | |
| **Total Cost Without Batteries** | **$0.92** | **Total Cost With Batteries** |
| 2 AA Alkaline Batteries and Holder (**xiii**) | $0.13 | $1.05 |
| 3 V Coin Cell Battery and Holder (**xiv**) | $0.06 | **$0.98** |

The table lists the costs for acquiring individual components in bulk of 10000 pieces. The LoCHAid is assumed has been created from the following: (**i**) a set of earphones (ModelGF-923, In-Ear, 3.5mm Connector, from Boluo Golden Fortune Electronic Manufacture Factory, www.alibaba.com, P/N 60249739970), (**ii**) a audio jack (1/4" 3.5mm PCB Mount Female Socket 5 pin, from Yueqing Daier Electron Co. LTD, from www.alibaba.com, M/N EJ-214M); (**iii**) 2 1000pF capacitors (SMD/SMT 1000 pF 50V Multilayer Ceramic Capacitor, from Part Rescue Technology, from www.alibaba.com, M/N VJ0603Y102KXACW1BC); (**iv**) 2 1 uF capacitors (SMD Ceramic Capacitor 1uF 50 V, from Shenzhen Yuzens Technologies Co. Ltd, from www.alibaba.com, M/N CL10A105KB8NNC); (**v**) a 15uF capacitor (250V 450vac 15uF polyester capacitor, from Shenzhen Weitaixu Cpacitors Co., Ltd., from www.alibaba.com, M/N cbb61 15uF run capacitor); (**vi**) a 5 kΩ Trim Pot Potentiometer (Cermaic Bourns Variable Resistor, from Changhoo Kennon Electronics Co. Ltd., from www.alibaba.com, M/N 3006P); (**vii**) a 6 pronged slide switch (Mini Slide switch, from A-Key Electronics Technology, from www.alibaba.com, M/N MSS-22D16); (**viii**) an electret microphone module (Utilising MAX9814, from Shenzhen Ronghai Electronics Co. Ltd, from www.alibaba.com, M/N MAX9814); (**ix**) a stereo 3.7 W amplifier (MAX98306 Stereo 3.7W Class D Amplifier, P/N MAX98306ETD+, from www.maximintegrated.com. P/N MAX98306); (**x**) a circuit board (Prototyping Universal Board PCB Double Sided 4 x 6 cm board, from Shenzhen Androw Technology Limited, D/C YC045-53, www.alibaba.com P/N 60529535100); (**xi**) 3D printed PLA casing is obtained in bulk (PLA plastic granules for 3D filament 3D material PLA plastic pellet, from Yasin, Guangdong China, from www.alibaba.com, M/N PLA pellets, JSC-310); (**xii**) 2 6.8 kΩ Resistors (Resistors 0.4 W 6.8 kΩ, from Shanhai Group Limited, from www.alibaba.com, M/N MMA02040C6801FB300). The LoCHAid can be powered by several types of batteries as long they deliver 3V; here, we present two forms—(**xiii**) 2 AA batteries (Entop 1.5V AA Carbon Zinc, from Suzhou South Large Batter Co., Ltd., www.alibaba.com P/N 60643508502) which needs a battery holder (2 AA 1.5V Battery Holder, from Yueqing Daier Electronics Co., Ltd., from www.alibaba.com, M/N BH5-2003); (**xiv**) or a coin cell battery (3V Lithium Button Cell, from Shenzhen Gmcell Technology Co., Ltd. P/N CR2032, www.alibaba.com, P/N 60251728326) which needs a coin cell holder ($0.03 (Black 3V Coin Button Holder, Yueqing Daier Electronics Co., Ltd., from www.alibaba.com, M/N BH2032-3). *All links and prices last accessed September 17, 2019.

nature allows LoCHAid to be replaced very quickly and cheaply if parts are damaged, resulting in a relatively easy-to-use OTC device. Labor costs are not considered in the price point, as the device is intended to be manufactured by the individual (see Discussion below). A personalisable (and potentially fashionable) custom case can be readily 3D-printed using other polymers than Nylon 12 (which is shown in Fig 1a) at potentially a slightly higher price point. However, other materials can be readily used for the case, including acrylic, cardboard, and foam. Given that most hearing aids and PSAPs cost around $4700 and $300 for a pair respectively, our device shows a reduction of cost by 99.98%.

## Electroacoustic analysis

The WHO Preferred Product Profile for hearing aid technology in low- and middle-income countries (LMIC) has recommendations for certain electroacoustic parameters [12]. The Consumer Technology Association (CTA) of United States, also established guidelines for electroacoustic parameters for OTC devices in wake of the 2017 FDA Reauthorisation Act [23, 24]. These parameters are OSPL 90, OSPL 60, Range of Frequency Response, Total Harmonic Distortion at 500, 1000, 1600 Hz at 70 dB SPL input, Equivalent Input Noise (EIN), and High Frequency Average (1, 1.5, 3 kHz). The values for LoCHAid were benchmarked by using an AudioScan Verifit (version 3.1; AudioScan, Dorchester, ON, Canada) machine that tested the aforementioned parameters in accordance with the ANSI (American National Standards Institute)/ASA S3.22-2014 standards (Fig 2a–2c) [27]. Table 2 compares the parameters of LoCHAid, against WHO Recommendations and CTA level. The frequency response curves for the LoCHAid are shown in Fig 2d.

The overall average gain for the frequency response curve is 15 dB SPL. The total harmonic distortion at 500, 1000, and 1600 Hz is very low at 1%, much less than the limits posed by WHO (8% at 500 Hz & 800 Hz, 2% at 1500 Hz), and CTA (5% at 500 Hz). The device itself has low interference with signal integrity, which is a necessary requirement for understanding speech accurately. The maximum OSPL 90 is much higher than the OSPL 90 @ 1 kHz, which denotes that the frequency response is skewed towards one end of the spectrum. Observing the high frequency averages (HFA), we see that the HFA (4, 5, 6 kHz) @ OSPL90 is 10 dB SPL higher than the HFA (1, 1.5, 3 kHz), which shows that the skewness of the response is directed towards high frequencies. The curves shown in Fig 2d highlight that the device is more selectable for high frequencies (> 2 kHz), and less selectable (< 1 kHz) for low frequencies. This selectivity towards high frequencies is necessary to treat ARHL, as hearing loss increases with frequency (S1 Fig in S1 File). The EIN of the device is 10 dB SPL higher than recommended from WHO PPP and CTA; however, we discuss the implications of this in the discussion section below (also see SI Section IV in S1 File). Overall, we successfully meet 5 out of 6 criteria as set out by WHO PPP and CTA [12].

## Simulated gain measurements against ARHL profiles

**Coupler gain simulations using the speechmap test.** After examining the electroacoustic characteristics of the LoCHAid, we explore how closely its gain measurements match a range of ARHL audiometric profiles. We compiled a total of 11 clinically averaged ARHL profiles based on age, gender, ear, and severity from previous work (1994-2004, 2008) [2, 3]. These profiles are males and females between the ages of 60-69 for both left and right ears, males and females between the ages of 70-79 for both left and right ears, and three gender neutral ARHL profiles of increasing severity of ARHL denoted by X (mild), Y (moderate), and Z (severe). The clinically averaged profiles were taken from a total sample size of N = 1546 Females, 1345 Males that exhibit ARHL in the United States (S1 Fig in S1 File).

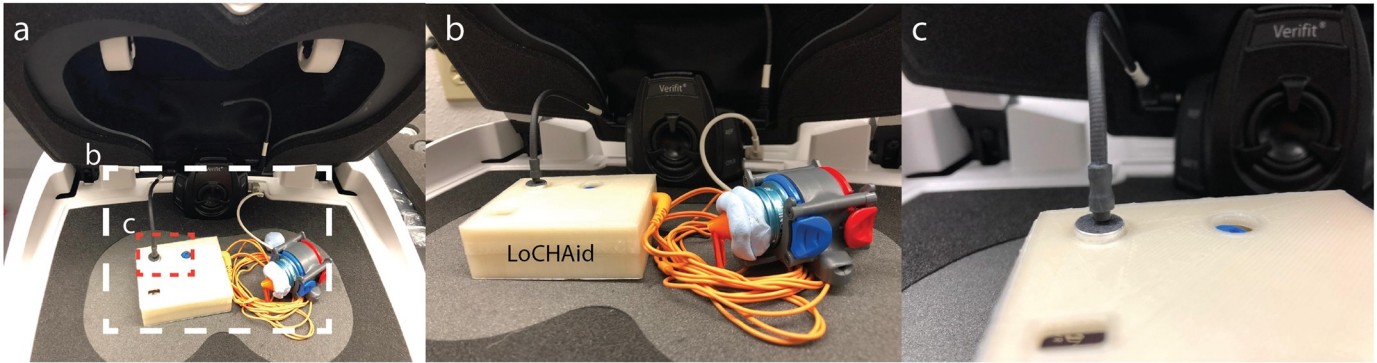

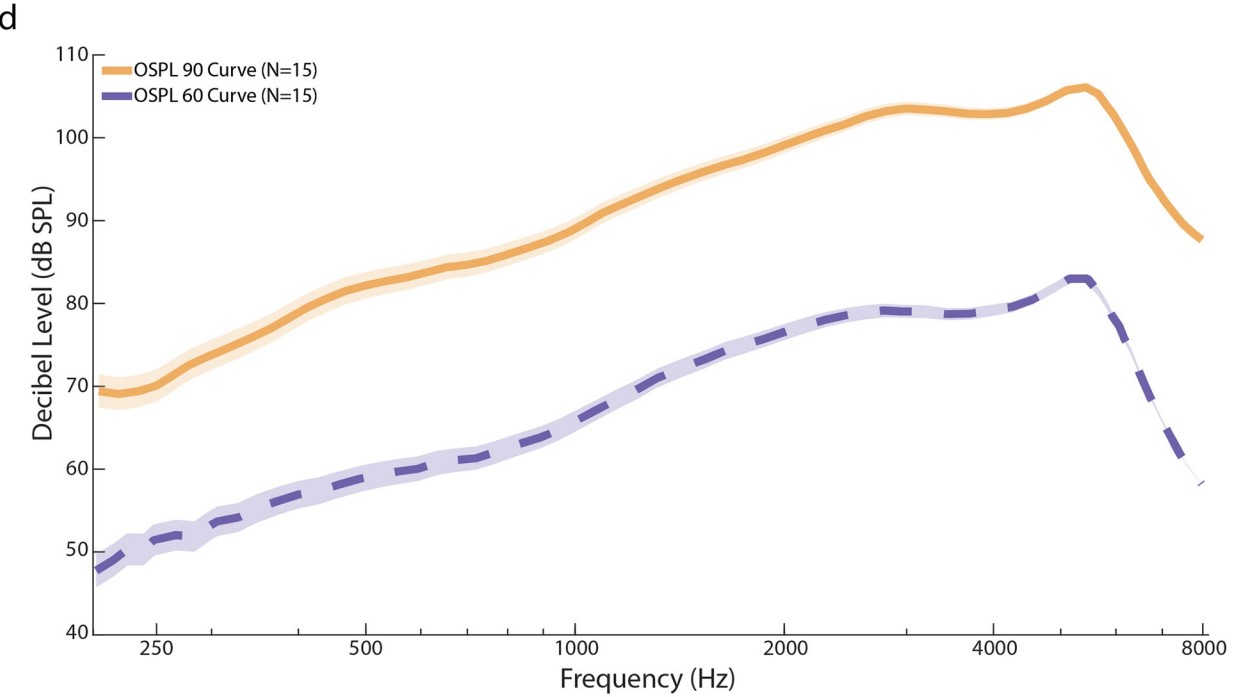

**Fig 2. Electroacoustic parameter testing setup and results. a**. A view of the device setup in the test-box. **b**. This image shows the setup of the device inside the AudioScan Verifit Chamber for testing. The external output of the headphones is placed with putty onto a blue 0.2 cc-coupler which is then attached to the instrument receiver module. **c**. This shows the placement of the AudioScan speaker output within 1 mm relative to the microphone input of the LoCHAid.**d**. The graph shows the OSPL 90 and OSPL 60 curves for the device (N = 3 devices, n = 5 trials per device). There is less amplification in the lower frequencies (< 1 kHz), and more amplification in the upper frequencies (> 1 kHz).

Speechmap measurements help show how closely the gain of the hearing aid at different frequencies matches the estimated gain required for ARHL audiometric profiles. The estimated gain for different audiometric patterns at different frequencies is governed by different hearing aid fitting algorithms. We chose the NAL-NL2 method, the current industry standard, which takes into account gender, age, and language [28]. The frequency targets are generated at 250, 500, 750, 1000, 1500, 2000, 3000, 4000, 6000, 8000 Hz, giving a total of 10 frequency targets.

Speechmap undertakes this simulation of NAL-NL2 targets based on an International Speech Test Signal (ISTS). ISTS is a mixed audio signal representing average speech at different frequencies and languages [25]. Three input sound levels for the signal were considered: 55 dB SPL (whispering level), 65 dB SPL (conversational level), and 80 dB SPL (loud level).

**Table 2. Electroacoustic parameter results and comparison.**

| Electroacoustic Parameters | WHO Recommendation | ANSI CTA-2051 | LoCHAid | Met? |
|---|---|---|---|---|
| Max OSPL 90 | 100-130 dB SPL | <120 dB SPL | 107 dB SPL | Yes |
| OSPL 90 @ 1kHz | 90-124 dB SPL | NS | 90 dB SPL | Yes |
| Average OSPL 90 | NS | NS | 96 dB SPL | |
| Average Gain | NS | NS | 15 dB SPL | |
| Total Harmonic Distortion @ 70dB SPL Input | 500 Hz <8%<br>1000 Hz <8%<br>1600 Hz <2% | 500 Hz <5% | 500 Hz = 1%<br>1000 Hz = 1%<br>1600 Hz = 1% | Yes |
| Equivalent Input Noise | <30 dB SPL | <32 dB SPL | 40 dB SPL | No |
| Range of Response and Smoothness | 200-8000 Hz<br>Smoothness—NS | 250-5000 Hz<br>Smoothness<br>No sharp peaks | <200->8000 Hz<br>Smooth | Yes |
| Battery Life | 2-3 Weeks | NS | 20 days<br>(with 2 AA batteries) | Yes |
| HFA (1, 1.5, 3 kHz) @ OSPL 90 | NS | NS | 93 dB SPL | |
| HFA (4,5,6 kHz) @ OSPL 90 | NS | NS | 103 dB SPL | |

The table lists the ANSI Parameters (OSPL 90, OSPL 60, Total Harmonic Distortion, High Frequency Average, Average Gain, Max OSPL 90) that were tested on the LoCHAid, the WHO Recommendations Preferred Product Profile (PPP) for the device, the ANSI/CTA-2051 recommendations, and the results from the LoCHAID, and whether the targets were met or not for both sets of recommendations [12, 27]. The device is able to meet all the targets except for Equivalent Input Noise. See Discussion in main text about EIN. *NS = Not Specified.

The response of the LoCHAID with full open volume against the targets of all 11 profiles is shown in Fig 3. To determine goodness of fit, we adopted a Strict and Loose Criteria that has been used previously by other researchers [11, 17, 19, 29, 30]. If the response of the device is within 5 dB SPL of the target, then it fits under Strict Criteria, while a response within 10 db SPL is used for the Loose Criteria. Under the Strict Criteria, all 11 profiles match only 10% of the targets, and 64% of the profiles match 50% of the targets. Under the Loose Criteria, 64% of the profiles match 90% of the targets, and all 11 profiles match 50% of the targets. The results reveal that the LoCHAid is a good fit to most profiles. However, not all the profiles are fitted equally well and the response of the device is too high to fit the milder ARHL profiles, such as Females in the 60-69 age range. To better fit the milder profiles, our data suggests to use the LoCHAid at a lower volume setting (-5 to -10 dB SPL). The reader is referred to S2 Fig in S1 File for quantification of fits for each profile, S9-S79 Fig and SI-SLX Tables in S1 File for individual profile targets and responses.

**Real-ear gain simulation using the G.R.A.S KEMAR manikin.** AudioScan, although reliable, measures the gain via a 0.4 cc wideband coupler, and is not the best representation of a real human ear. Hence, to obtain a more accurate and precise measurement of a real-ear, we used a G.R.A.S Knowles Electronics Manikin for Acoustic Research (KEMAR). This manikin is designed to anatomically resemble a real human ear as close as possible, and hence provides a real-ear simulation. The device was attached to the KEMAR manikin as shown in the inset of Fig 4a. The ear buds were placed into the ears, and an ISTS signal of 65 dB SPL was played.

Fig 4a details the targets and response for Males 60-69 Left and Right Ears. Under the Strict and Loose Criteria, 70% and 90% of the targets are matched, respectively, indicating an overall good fit for this ARHL audiogram. The results for all 11 ARHL profiles are shown in Fig 4b. Under the Strict Criteria, all 11 profiles match 50% of the targets, and 64% of the profiles match 70% of the targets. Based on Loose Criteria, 70% of all the profiles match 90% of targets, and all the profiles match 80% of the targets. The improvement in Strict Criteria matched

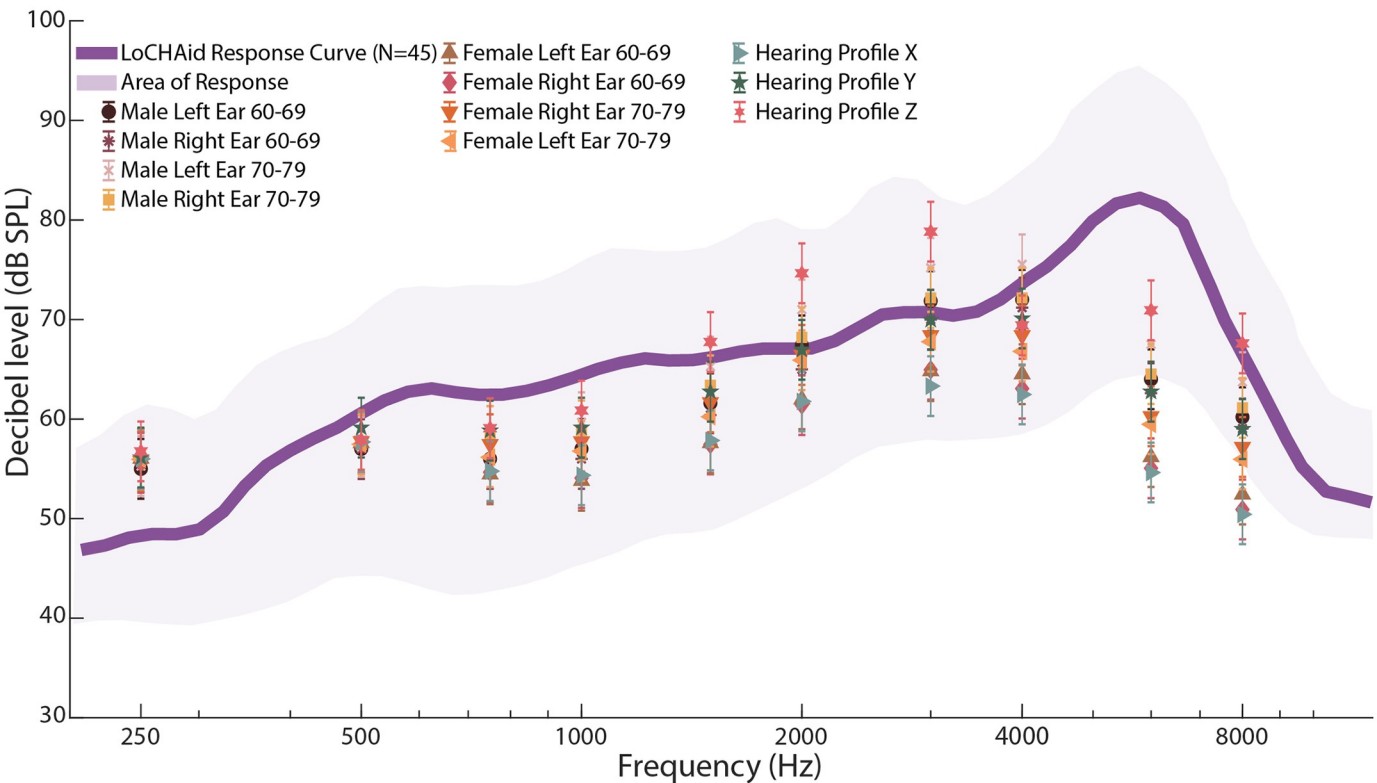

**Fig 3. Audiometric fitting results by speechmap.** The graph shows the NAL-NL2 targets for 11 profiles. The purple line is the average response of the device on full on gain (no volume reduction) in response to ISTS 65 dB input; the shaded area of shows the range of response of the device to the input. The targets have a standard error of 3 dB SPL, which are shown in the error bars. The objective is for the purple line to go through the targets for the device to be fit to the profile. The device well incorporates the range of targets in its area of response, and the average response is well within 10 dB of the targets except for 6000 Hz. The data is taken from N = 3 devices, n = 15 trials. The reader is referred to S2 Fig in S1 File for quantification of fits for each profile, S9-S79 Fig and SI-SLX Tables in S1 File for individual profile targets and responses.

targets from Speechmap to KEMAR for all the profiles is from 10% to 50%, and the improvement for Loose Criteria is from 50% to 80%. Both these improvements show that the device is very well fitted to the profiles. We note that 50% of all missed targets lie at low frequencies (250 Hz, 500 Hz) as the device shows very low gain at low frequencies (< 750 Hz). This is desirable as many individuals with ARHL often tend to report hearing echo of their own voice, and also hearing low noise such as 'refrigerator noise and humm' (100-200 Hz), which can be distracting [16]. The reader is referred to S2 Fig in S1 File for quantification of fits for each profile, S9-S79 Fig and SI-SLX Tables in S1 File for individual profile targets and responses.

## Discussion

### Potential advantages of LoCHAid platform

We designed the LoCHAid to be as affordable as possible at 98 cents (< $1, which is less than price of a bottle of water or a cup of coffee). A WHO guideline states that a hearing aid should be no more than 3% of the gross national product, per capita, per hearing aid [31]. Using current World Bank Figures, a hearing aid in order to be affordable has to be within $1614 for United States, $62 for India, $10 for Ethiopia [32]. For low-income, lower-middle income, and low and middle income countries, the affordable price is $20, $67.77, $135, respectively. Our device clearly meets this criteria [32]. Additionally, the LoCHAid is at most 20% per capita

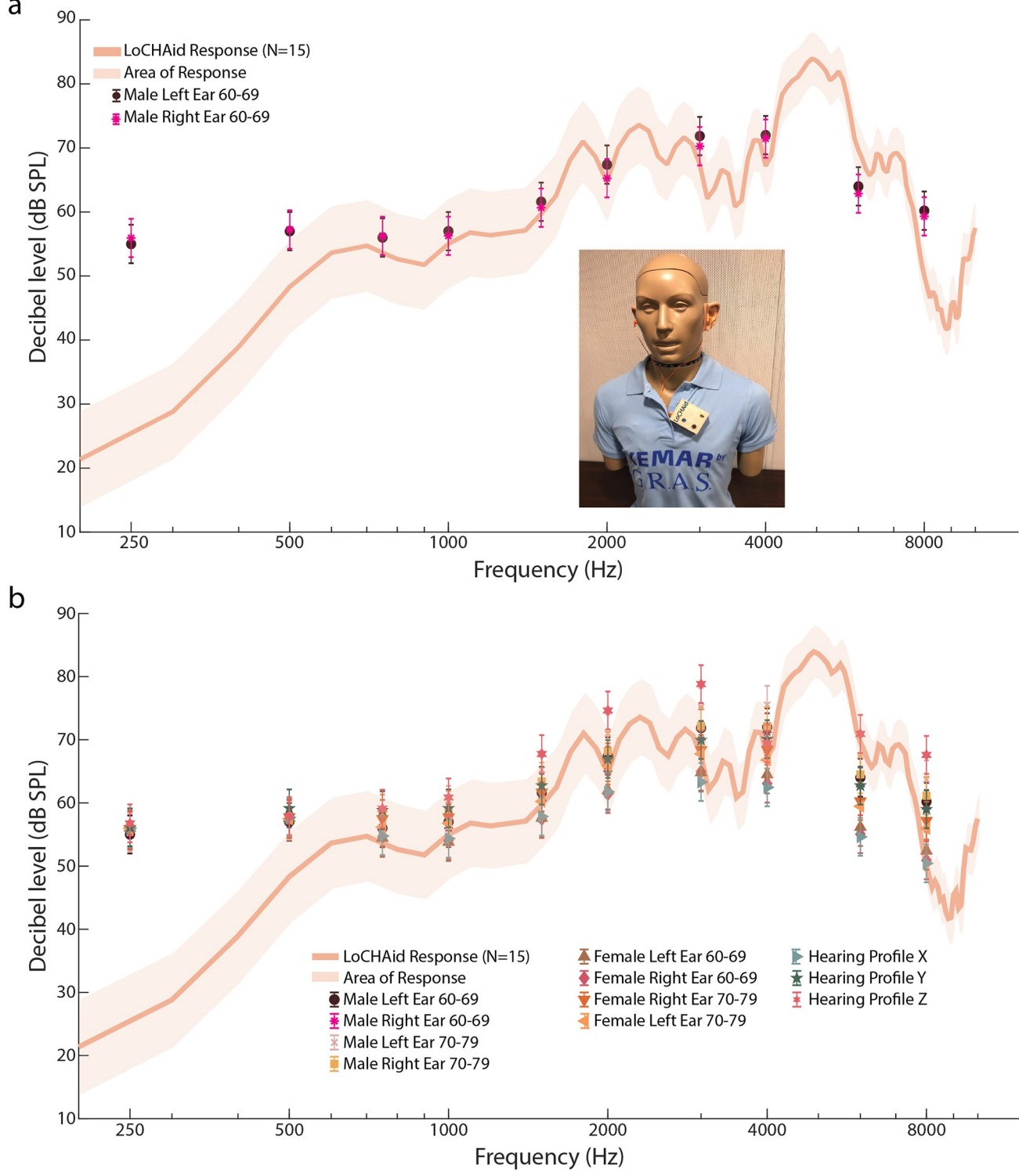

**Fig 4. G.R.A.S KEMAR audiometric fitting results. a**. The graph shows the LoCHAid KEMAR Response with NAL-NL2 targets from profiles Males 60-69 Left and Right ears. The solid line shows the average response of the KEMAR real-ear (N = 1 device, n = 15 trials); the shaded area represents the standard deviation of the response. The NAL-NL2 targets for the profiles from SI Fig in S1 File are shown as well to see how well the KEMAR response fits the targets. The fits are better than the test-box simulations in Fig 3. Overall, the graph shows that the device very well fits the profiles within 5 dB SPL, except at lower frequencies. However, at lower frequencies, it is better to have less gain, as there is user complaint of hearing background noise. The inset image shows the setup of the LoCHAid with hearing buds on the G.R.A.S KEMAR Manikin of testing. **b**. The graph shows the NAL-NL2 targets for all 11

profiles with the response of the LoCHAid. The response is better matched to the targets in the KEMAR simulation as compared to the test-box Speechmap simulations, under both Loose and Strict Criteria—see Discussion in main text. The reader is referred to S9-S79 Figs and Tables I-LX in S1 File for individual profile targets and responses.

annual health expenditure ($5–$50) for LMIC [20, 21]. For upper middle- to high-income countries, it is less than 1% of annual health care expenditure per capita ($1000-$3,000) [20, 21]. We have accomplished this by leveraging off-the-shelf components, mass produced electronics, and printed circuit boards. The lack of specialised electronic components such as digital sound processors and wires all help to not only make the device affordable, but also minimal in the number of components and design.

We understand that integration of technology with elderly patients is an ongoing challenge, and the ease of use is a key indicator in successful adoption. We have designed our device to be accessible for elderly individuals with mild to moderate ARHL. The device is body-worn rather than behind-the-ear (BTE) or in-the-ear (ITE). The design is geriatric friendly; many elderly patients have trouble handling the small in-the-ear, and BTE and ITE hearing aids, especially those with limited dexterity as a condition of arthritis [17, 33, 34]. The larger model reduces the likelihood of elderly patients misplacing the device, and facilitates the use of slightly larger domestic lithium-ion batteries. Since batteries are an additional cost, we opted to use lithium-ion batteries. Lithium-ion batteries enable longer usage times than zinc-ion batteries, and do not require trips to costly specialised battery markets, which often do not exist in LMIC [31, 35, 36]. Previously the cost burden of batteries has been notably addressed by solar technologies such as solar panel rechargers by groups such as Solar Ear [37]. We note that our design is indeed compatible with this philosophy and an off-the-shelf solar charger can be readily employed to charge the lithium ion battery as shown in S4 Fig in S1 File. The combined component cost for the solar panel, adapter, lithium-ion battery, and LoCHAid is still only $26.22 [38]. Thus, the hidden annual cost burden of non-rechargeable batteries is also reduced.

We have made the manufacturing and distribution of the LoCHAid accessible as well. Currently, the distribution methods of hearing aids are not direct to consumer [8, 31, 34]. Hearing aids are sold by specialists who are typically audiologists; ear, nose, and throat physicians; and licensed hearing-aid specialists [8, 34]. Practices such as bundling, limited selection of devices, and vertical integration of independent audiological clinics by hearing aid companies, have created barriers to access [34]. Our device circumvents the need for specialised dispensers through its minimalist design, which can be marketed through OTC. We have not only made the device OTC, but also do-it-yourself. The current PCB configuration is through-hole as it is the easiest to solder upon when manufacturing. Like other free and open source hardware (FOSH), our open-source device empowers local communities to be involved in every step of use of the device, from its creation and construction, to repair and disbursement of devices to those in need. Effectively, the open-source nature of LoCHAid makes it accessible for communities to create their own supply chain logistics, which was not addressed in previous work in hearing aids for LMIC [39]. Such an approach to combine appropriate technology with a local support base is essential to meet the needs of LMIC, as there is a chronic shortage of trained support personnel for hearing aids [31, 40].

The LoCHAid represents an opportunity to change the value proposition of hearing aids. In European countries such as the United Kingdom, where hearing aids are fully or partially covered under governmental health programs such as National Health Service, uptake remains low at 30% too [8, 13, 34]. Social stigma is one of the barriers; however, that may be changing with the arrival of an aging population that has grown more comfortable with technology and have a desire for more fashionable, robust, and better hearing technology [1, 34]. With

LoCHAid, individualisation of the device is just a matter of time. Like owning different pairs of glasses, one can create different 3D-printed casings and designs make it fashionable to one's desire. It creates an opportunity to induce a perception shift, where hearing aids are not seen as a hindrance, but an extension of one's personality.

## Potential shortcomings and future work

A key assumption that we make about the LoCHAid technology is that it can be easily constructed by the user. Since ARHL affected individuals are elderly people, it will be difficult for them to make their own device due to the need of handling the small electronic components that are part of the LoCHAid. Secondly, the user may find it difficult to gather the requisite parts for the LoCHAid from different sellers and markets. Thirdly, there is a lack of trained individuals in LMIC who can work with hearing technologies. We realise that this initially requires the LoCHAid to be shipped as a finished device, before established infrastructure for distribution, construction, and repair centers are setup for the device. The cost may subsequently be above the 1 dollar price point of components due to these additional costs of labor, assembly, and distribution. As an example, we obtain quotes from a mass manufacturer for our device (MacroFab, www.macrofab.com) and receive a total cost of $2.40 (for 10,000 units), that includes components and assembly, but not shipping. We are currently exploring further local manufacturing options and distribution networks, which will be the focus of future work. A successful example of such an idea is the $1 microscope (FoldScope), which is now commercially delivered at similar low-cost price points [41].

A potential shortcoming of our device is its large size and form factor—this may not be convenient or appealing to all consumers. To address this, we have developed a smaller prototype where the device is 1.05 x 0.81 in (S7-S8 Figs in S1 File), a reduction of more than 93% of the current size. However, at this small footprint, this device is very difficult to assemble by hand due to requisite surface mounting of the MAX9814 and MAX98306 chips on the board. The cost of manufacturing drives the price point higher than $1; at MacroFab, assembly and labor is quoted at $7.80 per unit for 10,000 units. Thus, there is a sharp trade-off between size and cost as the need for third-party assemblers arises.

We have conservatively estimated the device lifetime to be 1.5 years, which is considerably less than an ideal lifetime of 5 years. However, with the low cost of manufacture, we anticipate that even with replacement costs, the LoCHAid will still be affordable. We have not conducted failure tests or lifetime tests of the LoCHAid to a considerable extent in this paper. A part of our future work will be seeing the effectiveness and usability in different environments and locations throughout the world.

The electroacoustic analysis shows that the LoCHAid has high frequency gain necessary for ARHL and meets most of the preferred product profile for hearing aid technology suitable for LMIC set out by the WHO. The one characteristic that is deviant from the standard is the EIN. Other researchers have noted that EIN is a measure that is most frequently out of specifications [42, 43]. In a recent study, four most widely used hearing aid models were tested which had an average EIN between 27 to 34.5 dB SPL [43]. Thus, we anticipate that the relatively high EIN of 40 dB SPL may hinder speech perception in some users, especially those with relatively mild hearing loss. The EIN can be reduced in future versions of the LoCHAid, potentially at an increased cost. We also note that the construction environment is key factor in EIN; a poor local environment for construction can lead to higher EIN as hardware placement and quality can be affected.

The LoCHAid is currently a one-size-fits all approach and is non-programmable, which is a disadvantage as ARHL individuals may have a wide range of hearing loss profiles, and our

current tests are performed against averaged audiograms. In future iterations, we are developing improvements to include programmability through a variable potentiometer in the second-order high-pass filter instead of keeping it constant at a fixed frequency cutoff. The variable potentiometer will be able to change the frequency cutoff, and hence change the frequency response. A further step is to implement digital signal processors (DSPs) and software to provide a wide programmable scale; however, at increased cost. We understand that this is a challenge in finding the right balance between cost and programmability, and we are currently exploring this in tandem with finding a lower EIN for the device.

The lab measurements presented here are a first step. Further translation and clinical work is necessary to evaluate the individual benefits and outcomes provided by the LoCHAid device. The use of earbuds as the receiving sound transducer is not the most ideal solution as it can potentially lead to an occlusion effect while talking. Additionally, the signal that we tested our device with was the ISTS signal, which utilizes English, German, French, and Spanish as its main components. For the vast majority of ARHL afflicted people who are in LMIC, these may not be their preferred spoken languages [1, 8]. To address all these shortcomings (which are beyond the scope of this current paper), we have established clinical collaboration with audiologists in India and Malawi, and are currently in the process of obtaining clinical evaluation of our device in a LMIC-context.

## Conclusion

Despite these shortcomings and limitations, here we offer a proof-of-concept low-cost hearing aid that has potential to address the challenges in accessibility and affordability for hearing aids. In the United States, hearing technology regulations are being reconsidered in the wake of the FDA Reauthorisation Act of 2017 [15]; our device is perhaps in the right time period to facilitate discussions between citizens, policy makers and audiologists. Beyond the United States, in LMIC, where the growing burden of ARHL is a serious concern, the LoCHAid offers an opportunity to indeed bring 'hearing to the masses' [44].

## Supporting information

**S1 Video. Construction of the LoCHAid.** Video outlining construction of the LoCHAid with Lithium Ion Coin Cell Battery. Video speed has been increased to 15x; however, the average time of construction is 25 minutes. S6 Fig in S1 File shows the schematic of the hearing aid.
(MP4)

**S2 Video. Preparing earphones for testing.** Video outlining how to properly set up device earbuds for audiological testing in AudioScan Verifit with 0.2-cc coupler.
(MP4)

**S3 Video. Water depth testing of device.** Video showing device after being submerged in 6cm of water, and shows that it still is in working condition. A still photo is shown in S5b Fig in S1 File.
(MP4)

**S4 Video. Drop testing device.** Video showing device after repeated drop tested (n = 10) from a height of 1.5 meters, and showing that the device is in working condition. A still photo is shown in S5a Fig in S1 File.
(MP4)

**S1 File.**
(PDF)

## Acknowledgments

We would like to thank A. Dubey and S. Sekhar for initial LoCHAid prototype designs; Dr. A. Dockens, Dr. E. Burns of Lamar Audiology for assistance in audiological testing; Dr. R. A. Robinson Jr at the Georgia Tech School of Electrical and Computer Engineering guidance on filter design and frequency response; J. Eng for assistance on design of the custom PCB; Dr. P. S. Russo at Georgia Tech School of Materials Science and Engineering for assistance in testing device; Dr. Bradley McPherson and Bhavisha Parmar for helpful suggestions and advice; and finally the Bhamla Lab for invaluable feedback.

## Author Contributions

**Conceptualization:** Soham Sinha, M. Saad Bhamla.

**Data curation:** Soham Sinha.

**Formal analysis:** M. Saad Bhamla.

**Funding acquisition:** M. Saad Bhamla.

**Investigation:** Urvaksh D. Irani, Vinaya Manchaiah, M. Saad Bhamla.

**Methodology:** Vinaya Manchaiah.

**Project administration:** M. Saad Bhamla.

**Resources:** Vinaya Manchaiah, M. Saad Bhamla.

**Supervision:** M. Saad Bhamla.

**Validation:** M. Saad Bhamla.

**Visualization:** Urvaksh D. Irani, M. Saad Bhamla.

**Writing – original draft:** Soham Sinha, Urvaksh D. Irani, M. Saad Bhamla.

**Writing – review & editing:** Soham Sinha, M. Saad Bhamla.

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
