## [Decision Letter · Decision Letter 0]

14 Jul 2020

PONE-D-20-05119

LoCHAid: An ultralow-cost hearing aid for age related hearing loss

PLOS ONE

Dear Dr. Bhamla,

Thank you for submitting your manuscript to PLOS ONE. After careful consideration, we feel that it has merit but does not fully meet PLOS ONE’s publication criteria as it currently stands. Therefore, we invite you to submit a revised version of the manuscript that addresses the points raised during the review process.

We look forward to receiving your revised manuscript.

Kind regards,

Pedro de Lemos Menezes, Ph.D

Academic Editor

PLOS ONE

Journal Requirements:

Reviewers' comments:

Reviewer's Responses to Questions

**Comments to the Author**

1. Is the manuscript technically sound, and do the data support the conclusions?

Reviewer #1: Yes

Reviewer #2: Yes

2. Has the statistical analysis been performed appropriately and rigorously? 

Reviewer #1: Yes

Reviewer #2: N/A

3. Have the authors made all data underlying the findings in their manuscript fully available?

Reviewer #1: Yes

Reviewer #2: Yes

4. Is the manuscript presented in an intelligible fashion and written in standard English?

Reviewer #1: Yes

Reviewer #2: Yes

5. Review Comments to the Author

Reviewer #1: This interesting m/s describes a proof-of-concept project -- the development of an "ultralow-cost" hearing aid, particularly developed for individuals living in resource-challenged regions. The authors convincingly show that it is possible to construct, using off-the-shelf components, a very low cost device that can provide basic amplification for adults with a wide variety of hearing loss configurations. The paper is of interest not only because of the affordable nature of potential devices built using this strategy but also because the authors point out some of the barriers to widespread take-up of such a device. I feel that, with some minor revisions, the m/s would be of value to the audiological scientific community.

There are a number of points where the authors should consider revisions. These are:

The Abstract claims the h/aid "costs only 98 cents (< $1) to mass manufacturer". However, the text states that mass manufacture would indeed cost substantially more than this. The authors should take care not to "over-sell" their proof-of-concept device in the Abstract. The first sentence is rather odd and should be altered -- do the authors mean that hearing aids are the primary tool in non-medical rehabilitation for persons with hearing loss? The "audiological market" as is written is a very vague and wide term and should be avoided. Consumer Technology Association for Hearing Aids. I don't understand why Hearing Aids is capitalized.

Introduction

p.2 Hearing aids are the "most frequently used" what in rehabilitation? Something missing. Same for "public policy" .. public policy what?

"..have been reported ... charateristics and .." is redundant and can be deleted.

p. 3 "supported" seems wrong in this context ... advocated would be more appropriate. line 70 .. both coupler and real-ear simulator measures were conducted through a Verifit Speechmap hearing aid analyzer. Or was another device used to measure the h/aid output from the KEMAR manikin? This sentence needs correction.

p. 4 better if "requires few soldering points"

p. 5 line 143 The THD statement is misleading. WHO only allows 8% at certain low frequencies, not as an overall figure. This statement needs correction.

p. 7 "strength 65 dB SPL" is rather unscientific. Better if "an ISTS signal of 65 dB SPL was played".

p. 9 To avoid over-generalization besterr to state "which often do not exist in LMIC". Line 259 Why the "($140)" in the text? Seems not reason for this to appear.

p. 10 should be "mass manufacturer" [singular not plural].

p. 11 line 317 better if "a lower EIN for the device".

References

Ref 3 should be "Beaver Dam"

Ref 4 should be Journal of Neuroscience - delete "The" and "the official..."

Ref 5 has no journal pages numbers

Ref 15 National Academies of Sciences, Engineering, and Medicine should be capitalized.

Ref 20 World Health Organization is duplicated.

Ref 36 "McPherson" is incorrectly spelled.

Ref 37 No publication etc. details are provided.

Ref 38 No publication etc. details are provided.

Figures 3 and 4b are almost impossible to read. I suggest creating multiple figures, one for male data, one for female data and one for hearing profile X, Y, Z data.

Table II legend Need to write "PPP" in full

Throughout the m/s there were times when my PDF file showed missing spaces between letters, such as page 2, line 43 "... (e.g.,lack of...". Check the m/s thoroughly for such typos.

Reviewer #2: - Reposition the METHODS section, which should be after the introduction;

- The video mentioned in line 98 (Video 4) should show the impact of LoCHAid from different perspectives;

- Line 116: I suggest a sub-topic on the cost of the device;

- Line 289: About the smaller prototype, I suggest a figure that shows the final device.

6. PLOS authors have the option to publish the peer review history of their article (what does this mean?). If published, this will include your full peer review and any attached files.

Reviewer #1: No

Reviewer #2: No

---

## [Decision Letter · Decision Letter 1]

11 Aug 2020

PONE-D-20-05119R1

LoCHAid: An ultralow-cost hearing aid for age related hearing loss

PLOS ONE

Dear Dr. Bhamla,

Thank you for submitting your manuscript to PLOS ONE. After careful consideration, we feel that it has merit but does not fully meet PLOS ONE’s publication criteria as it currently stands. Therefore, we invite you to submit a revised version of the manuscript that addresses the points raised during the review process.

We look forward to receiving your revised manuscript.

Kind regards,

Pedro de Lemos Menezes, Ph.D

Academic Editor

PLOS ONE

Reviewers' comments:

Reviewer's Responses to Questions

**Comments to the Author**

1. If the authors have adequately addressed your comments raised in a previous round of review and you feel that this manuscript is now acceptable for publication, you may indicate that here to bypass the “Comments to the Author” section, enter your conflict of interest statement in the “Confidential to Editor” section, and submit your "Accept" recommendation.

Reviewer #1: (No Response)

Reviewer #2: All comments have been addressed

2. Is the manuscript technically sound, and do the data support the conclusions?

Reviewer #1: Yes

Reviewer #2: (No Response)

3. Has the statistical analysis been performed appropriately and rigorously? 

Reviewer #1: N/A

Reviewer #2: (No Response)

4. Have the authors made all data underlying the findings in their manuscript fully available?

Reviewer #1: Yes

Reviewer #2: (No Response)

5. Is the manuscript presented in an intelligible fashion and written in standard English?

Reviewer #1: Yes

Reviewer #2: (No Response)

6. Review Comments to the Author

Reviewer #1: The revised m/s meets almost all my initial review comments and now has improved readability and focus. The m/s covers an area that is under-researched and I commend the research team for their exploratory work.

My remaining comments concern minor points. Attention to these would further improve the final m/s:

page 4/16 better if just "... with the manikin inside."; inconsistent use of units in the paper - here and in S4 Video legend feet are used rather than metric units. Metric units should be used at all times throughout the paper.

page 5/16 mentions an estimated device lifespan of only 1.5 years. This issue should also be very briefly considered in the discussion.

Table 2, page 7/16. change "HZ" to "Hz" as in all other instances in the table.

page 10/16 makes the claim that the low cost device can be used in "screening for potential hearing loss". This is a puzzling statement because nowhere in the m/s is a device screening function mentioned. I think this goes well beyond the scope of the present research paper and should be deleted.

Some references still have problems: #4 should be just "The Journal of Neuroscience"; #15 and #19 just "JAMA"; #23, #24, #26, #27 and #34 are all incorrectly titled and need revision. #43 there should be no space before "?"

Reviewer #2: (No Response)

7. PLOS authors have the option to publish the peer review history of their article (what does this mean?). If published, this will include your full peer review and any attached files.

Reviewer #1: No

Reviewer #2: No

---

## [Editor Report · Decision Letter 2]

27 Aug 2020

LoCHAid: An ultralow-cost hearing aid for age related hearing loss

PONE-D-20-05119R2

Dear Dr. Bhamla,

We’re pleased to inform you that your manuscript has been judged scientifically suitable for publication and will be formally accepted for publication once it meets all outstanding technical requirements.

Kind regards,

Pedro de Lemos Menezes, Ph.D

Academic Editor

PLOS ONE

---

## [Editor Report · Acceptance letter]

4 Sep 2020

PONE-D-20-05119R2 

LoCHAid: An ultra-low-cost hearing aid for age-related hearing loss  

Dear Dr. Bhamla:

I'm pleased to inform you that your manuscript has been deemed suitable for publication in PLOS ONE. Congratulations! Your manuscript is now with our production department. 

Kind regards, 

on behalf of

Dr. Pedro de Lemos Menezes 

Academic Editor

PLOS ONE